# Integrating perspectives of patients, healthcare professionals, system developers and academics in the co-design of a digital information tool

**Annika Grynne** [1☯*], **Maria Browall** [1,2‡], **Sofi Fristedt** [3‡], **Karin Ahlberg** [4], **Frida Smith** [5,6☯‡]

1 Department of Nursing, School of Health and Welfare, IMPROVE, Jönköping University, Jönköping, Sweden, 2 Affiliated to Department of Oncology, Inst of Clinical Sciences, Sahlgrenska Academy, University of Gothenburg, Gothenburg, Sweden, 3 The Jönköping Academy for Improvement of Health and Welfare, School of Health and Welfare, Jönköping University, Jönköping, Sweden, 4 Institute of Health and Care Sciences, Sahlgrenska Academy, University of Gothenburg, Gothenburg, Gothenburg, Sweden, 5 Regional Cancer Centre West, Gothenburg, Sweden, 6 Department of Technology Management and Economics, Chalmers University of Technology, Gothenburg, Sweden

☯ These authors contributed equally to this work.
‡ MB, SF, and FS also contributed equally to this work.
* annika.grynne@ju.se

**Data Availability Statement:** All relevant data are within the manuscript.

## Abstract

### Background

Patients diagnosed with cancer who are due to commence radiotherapy, often, despite the provision of a considerable amount of information, report a range of unmet information needs about the treatment process. Factors such as inadequate provision of information, or the stressful situation of having to deal with information about unfamiliar things, may influence the patient's ability to comprehend the information. There is a need to further advance the format in which such information is presented. The composition of information should be tailored according to the patient's individual needs and style of learning.

### Method and findings

The PD methodology is frequently used when a technology designed artefact is the desired result of the process. This research is descriptive of its kind and provides a transparent description of the co-design process used to develop an innovative digital information tool employing PD methodology where several stakeholders participated as co-designers. Involving different stakeholders in the process in line with recommended PD activities enabled us to develop a digital information tool that has the potential to be relevant and user-friendly for the ultimate consumer.

### Conclusions

Facilitating collaboration, structured PD activities can help researchers, healthcare professionals and patients to co-design patient information that meets the end users' needs. Furthermore, it can enhance the rigor of the process, ensure the relevance of the information,

**Funding:** The project was funded by the Regional Cancer Centre West (RCC West), Sweden, the Chalmers Innovation Office, Sweden, Knut and Ragnvi Jacobsson Family Foundation, Sweden, Jönköping University, IMPROVE, Sweden. The funders had no role in study design, data collection and analysis, decision to publish, or preparation of the manuscript.

**Competing interests:** The authors have declared that no competing interest exist.

and finally have a potential to employ a positive effect on the reach of the related digital information tool.

## Introduction

It is not uncommon for persons diagnosed with cancer to experience challenges in their care [1], and these may relate to the provision of information [2]. Patients who have received some prior information about mediations related to their care tend to have more interactive discussions with the healthcare professionals, and have a more positive experience before, during and post-treatment [3]. Further, when stakeholders are involved in developing their own information material, it tends to become more relevant and readable [4]. Over 50% of all cancer patients are offered radiotherapy (RT) as a mode of treatment [5]. Because the high-tech RT environment is non-accessible to the public, it is an unknown environment for most patients diagnosed with cancer about to undergo RT. There is an association between a lack of information and knowledge related to the RT treatment procedures and increased levels of anxiety and feelings of being unprepared at the beginning of treatment [6, 7].

The primary purpose of patient information is to prepare and support the patients' and their families' needs throughout the complex RT process [6]. Providing accurate preparatory information to patients prior to RT will have a positive impact on patient-related outcomes and reduce misconceptions regarding treatment [8]. The traditional format comprises the healthcare professional providing information face-to-face, often in the form of one-way communication within a clinical setting, commonly reinforced by providing paper handouts [7]. Both the patient and the healthcare professional may have preconceptions of what type of information is needed and how it is best provided. The information provided to patients often outlines what the experts want the patients to know rather than considering what information the patients are able to comprehend [9]. Despite the provision of a considerable amount of information, patients still report a range of unmet information needs about the treatment process and how the treatment may affect their relationships with their family [1, 2, 10]. This may not so much relate to the amount of information provided but instead to the format and provision of the information.

Several factors may influence the patient's ability to comprehend health information. The ability may be associated with inadequate provision of information [3], e.g., the way in which the information is presented may go beyond the reading comprehension level of many adults, or the patient is presented with a high load of information over a short period of time. This signifies that there is a need to further advance the format in which the information is presented. There are a considerable number of initiatives providing diverse categories of information, however, few are specifically directed towards RT. The composition of information should be tailored according to the patient's individual needs and style of learning and involving patients as stakeholders can open new and more effective ways to ensure this [11]. In this article, we describe the use of participatory design (PD) methodology in the development of a new digital information tool directed towards patients who have been diagnosed with cancer and are receiving RT treatment.

### Developing digital information technology with stakeholders

Moving from involving patients and healthcare professionals as consultants in the later stages of the development of patient information material to engaging both groups together as active

participants in co-designing patient information material is a relatively new concept that is not commonly explored [6, 8]. Active involvement of these groups as stakeholders in co-designing patient information material will provide unique insights into their experience and needs. Further, it is more likely to result in information material that has a higher relevance, which is preferred to information material produced solely by the clinical experts [4, 12].

## Adapting patient information–health literacy

Patients bring their own unique experience to a healthcare encounter, and the healthcare professionals bring their clinical knowledge and experience [13]. The difficulties that patients have in assimilating generic patient information into their own personal and social contexts and experiences may be associated with low levels of health literacy (HL). In this article, HL involves the ability to obtain, understand, communicate, and act on information on health issues to promote and maintain health [14]. Mårtensson and Hensing [15] suggest that HL can be understood as a dynamic continuum. HL may be viewed as a context-dependent phenomenon that can be affected by the anxiety the patient experiences due to the stressful situation of having to deal with information about unfamiliar things [15]. Adequate HL may be achieved in various ways. Patients have different approaches in how they can best adapt patient information to their situation; hence, there is a call for comprehensive information to be tailored in a format best suited to the individual [4]. Adopting a person-centered approach, where the patient is viewed as a capable person beyond the diagnosis instead of placing the focus on the disease alone [13], will not only reduce misconceptions regarding RT [8, 16, 17] but also increase the patient's ability to make an informed decision regarding their treatment plan [18].

## Digital patient information and a person-centred approach

The emergence of digital technology provides innovative opportunities and can assist in providing person-centered information, enabling patients to make well-founded decisions [6, 19]. There are indications that patients prefer to be given information that they can access later for review at home and for sharing with others [2, 19]. Unlike traditional face-to-face information digital technology presents a range of alternatives enabling the patient to select the format of information best suited to them and where they can apply the information in an iterative manner. Rapid advances in digital technology generate both opportunities and challenges in providing patient information and there is a strong call for digitalized tools to be used in healthcare. However, it is important that the development digital tools are based on research that follows quality standards [20]. Furthermore, rigorous research evaluating the validity and efficacy of such tools are scarce [21]. Digital technology can, in an innovative graphical way, aid the preparatory information by enabling the patient to experience something previously unknown [2], in our case, the highly-technological RT treatment room, presented in 3D by applying Virtual Reality (VR)-glasses, or watching animated films in an information app. Employing VR technology presents an inherent potential in being interactive because its content is generated in real time, not ahead of time, which makes the experience even more immersive [22]. The VR-technique allows patients a simulated experience of viewing the environment prior to their first visit to the RT department. Furthermore, digital technology, enables the patient to have a simulated experience as well as access the information for review in the comfort of their home. It also allows the distribution and sharing of information with their family and friends [2, 19]. Although digital technology is depicted as "empowering" for both patients and staff, there are challenges that need to be considered to avoid a digital divide, as some patients, particularly those age 75 years and older, still lag behind younger persons in its adoption [23]. e-Health literacy is defined in similar terms as HL, in that it refers to the

ability to seek and appraise health information from electronic sources and to apply the gained knowledge to address health problems [24]. This insecurity may hinder patients from being confident enough to use a digital information tool involving mobile devices. This may be referred to as mobile health (mHealth) literacy, viewed as a subset to eHealth literacy, and refers to the use of mobile technologies for medical and public health practices [25].

While VR is commonly used for distractive interventions to relieve pain and distress during medical procedures [26, 27] there is a growing interest in the use of VR-interventions in managing cancer-related symptoms [21]. Furthermore, it has been found that VR-interventions in relation to patient information has a positive effect on understanding of the RT process while reducing anxiety [28, 29]. While digital technology is increasingly integrated into clinical services, it is not known from research how the VR-technology can be tailored to present a convenient approach to meet each individual person's style of learning to adapt patient information related to RT-treatment. This article is part of a larger prospective Randomized Control Trial (RCT) study and constitutes of a foundation and groundwork for the project. While the efficiency and relevance of the digital tool will be tested in the RCT, within this article, we aim to describe the PD process in developing a digital information tool with VR-technology to support persons diagnosed with cancer and increase their perception of being prepared for their planned RT treatment. It is worth to note that although PD is applied it is nothing that we will measure within the scope of this article. This is original research that is more descriptive of a kind.

## Method

### Participatory design

Under the umbrella of co-design, we followed the specific PD methodology developed by Spinuzzi [30]. PD is an approach that seek to actively involve all stakeholders in the design process and is often used when a technology designed artefact is the desired result of the process [31]. With the ambition to create a digital information tool with VR-technology for persons diagnosed with cancer, we involved relevant patient groups as well as healthcare professionals as stakeholders in the process. Adopting a dynamic approach to HL [15] alongside a person-centered approach [8], we aspired to produce a format of information, tailored in line with the patient's individual style of adapting information. Applying participatory activities enables the clinicians, researchers, and patients to co-design a tool that meet patients' needs [4, 8, 32]. Ensuring that the digital information tool is relevant and user-friendly for the consumers, and that it is based on scientific evidence and experience-based knowledge, requires the involvement of several stakeholders [8, 31, 32]. The stakeholders' participation can range in activity; in our case, applying PD, they were involved in the design process as well as assessing and testing the feasibility of the digital information tool in a pilot study. Other activities can include taking on more of a consultant role, where the stakeholder is asked to comment on pre-designed material [11, 30]. To be transparent about the involvement of the stakeholders, we have applied a specified work process inspired by a protocol developed by Elwyn [33].

### Setting, participants and procedure

This project took place in a RT-department in a University hospital in Sweden. The project described within this article commenced 2017, and the pilot study was undertaken spring 2018. The design of this study is of descriptive origin and to enable a better flow of the text, some of the methods and results are sometimes presented unitedly. With respect to ethical considerations, the study is part of a RCT approved by the Swedish Ethical Review Authority (DNR 2020–00170), and for the pilot study, (DNR 917 17), written informed consent

was obtained from the patients. For the different steps of the development process, the study was approved by the heads of the radiotherapy clinic; verbal consent was obtained from the participating members of staff and the Innovation Development Team through voluntary participation. A list of participants was provided by the head of staff to the researchers. No written consent was considered necessary since there were no personal or sensitive issues discussed.

The PD methodology, which involves three stages; initial exploration of work, discovery process, and prototyping [30] was, in relation to this article, combined with a protocol developed by Elwyn [33], which contains a variety of complementary steps in the work process (Fig 1). The development of the digital information tool emerged in the first four complementary steps, and reflections gathered in the co-design approach informed the development of the digital information tool in the last step (Step 5). A project management group was formed, consisting of three researchers: one with previous experience of PD (FS), one with experience of the RT-process (KA), and one with experience of the cancer care process (MB). All three had experience of both qualitative and quantitative research involving the intended population and were well established within the field of cancer nursing. A professional innovation development team (IDT) i.e., system developers with expertise in gamification within healthcare, were involved and active throughout the process of developing the digital information tool. Healthcare professionals, administration staff, and persons diagnosed with cancer undergoing RT acted as stakeholders in the initial need's assessment (Step 1). For Steps 2 and 3, an advisory group, including a consolidated group of registered nurses and managers, were invited to review the accuracy of the text. A team member of the IDT had personal experience of a cancer diagnosis and RT treatment and was therefore able to take on an additional role within the project to act as a valued patient representative through the process. The project management group members had executive and editorial control as well as responsibility for making the final decisions.

**Initial exploration of work.**   Step 1—One of the researchers in the project management group (KA) had a role as clinical associate professor at the RT department. Together with the manager at the RT department KA was able to do a strategic selection among the staff to ensure a broad sample of professionals participating in the workshop. The staff was informed about the research and following verbal consent invited to participate in the workshop for an initial assessment. In the workshop they described what they, in their specific role, experienced were essential to know from a patient's perspective due to commence RT treatment. Through a strategic selection the researcher (KA) invited patients undergoing RT treatment to participate in a patient survey. Additionally, in this step existing written patient information (booklets) were identified and examined. The project management team wanted a strategic selection of individuals who could contribute to the co-design and development of the digital information tool. It was important that healthcare professionals, administration staff, IDT staff and patients, all voices to be heard in accordance with PD methodology.

**Discovery process.**   Step 2—The IDT with extensive experience in health information innovations was contracted to produce the digital tool version 1.0.

Step 3—The relevance, comprehensiveness, feasibility, and usefulness of the digital tool version 1.0 was tested and assessed in a pilot study.

**Prototyping.**   Step 4—Two separate group interviews were conducted with health care professionals and staff from the IDT. A semi-structured interview with predesigned questions was used. The participants were asked to share their experience of working in co-design with researchers to develop a digital tool. A thematic analysis was applied.

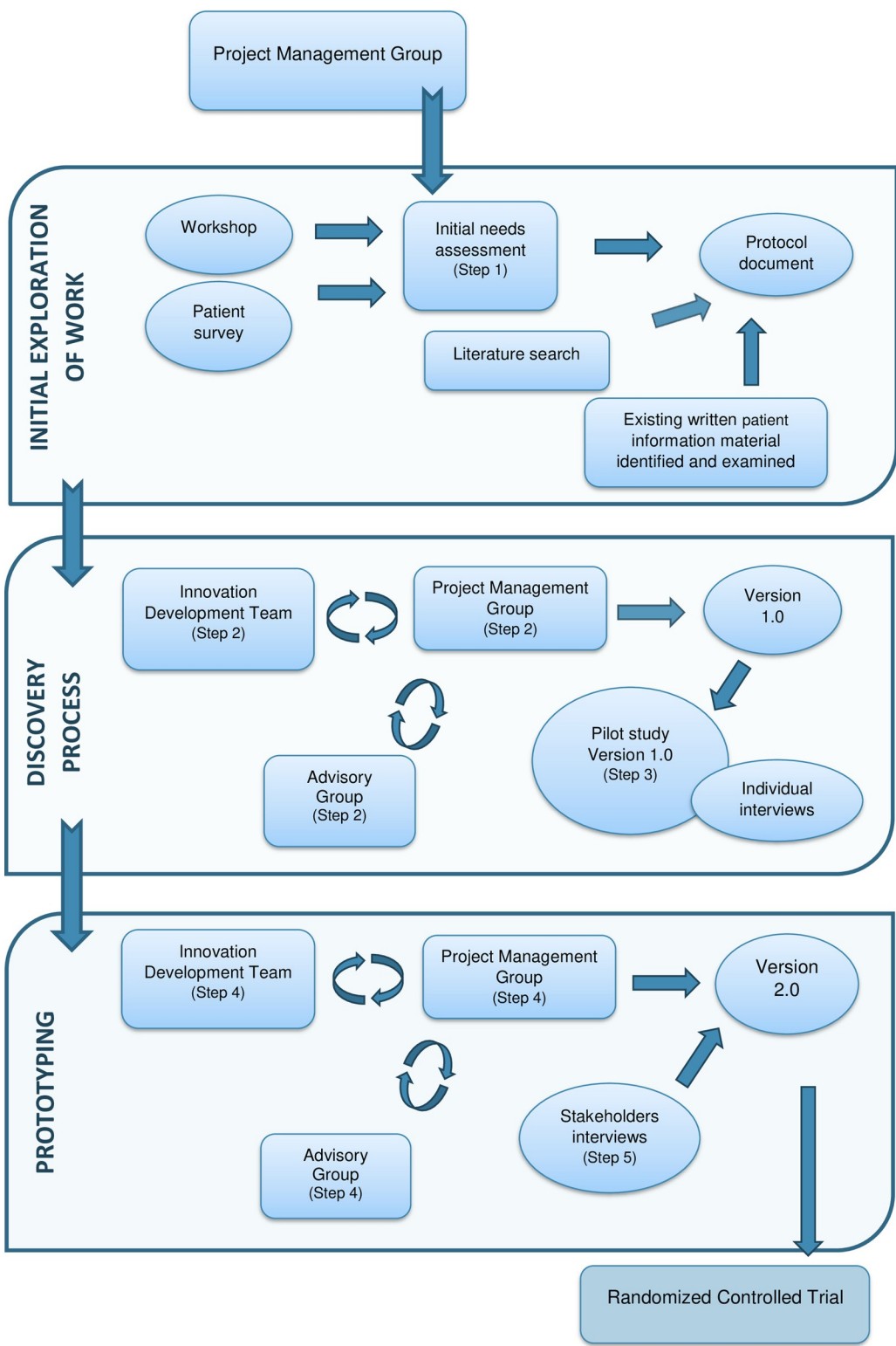

**Fig 1. Complementary steps in the work process.** The three stages relate to participatory design methodology combined with a variety of steps. The shaded areas illustrate steps of the development of the digital information tool (step 1–4) and reflection of the process (step 5).

# Results

## Initial exploration of work

**Initial needs assessment (Step 1).** An initial workshop took place where healthcare professionals and administrative staff working at a RT department at a university hospital discussed what information relating to RT, they perceived to be essential for patients to know before, during and after RT-treatment. Despite taking several approaches to recruitment, no patients found it possible to participate in separate workshop. Instead, following a strategic selection, a patient survey was administered to a group of 45 persons (13 women, 32 men) diagnosed with cancer and undergoing RT-treatment to measure their need of information. The survey was followed by a complementary individual short briefing at the time of their planned visit to the RT department. During the patient assessment stage, it became apparent that the patient's opinion of what kind of information they saw as essential stood in contrast to the information that the healthcare professionals and administration staff deemed to be necessary. Additionally, as part of the initial assessment, the project management group conducted a literature search with a specific focus on VR-technology as an information tool within cancer care. Finally, existing written patient information material consisting of evidence-based information in form of booklets with focus on cancer and RT used at the hospital was identified and examined. The existing written patient information material was developed by healthcare professionals at the RT department following an evidence synthesis of the literature. The information gained from reading the existing patient information together with the results from the workshop and patient survey, constituted the protocol document, which also describes the design of the project, the rationale, and the process for the development of the digital information tool [34].

## Discovery process

**Project management group & IDT (Step 2).** The IDT was contracted to produce the digital information tool version 1.0. The tool eventually comprised two separate applications (apps) for mobile devices: 1) a VR-app, presenting a virtual guided tour of the RT-department with a voice-over to describe 360-images to create a sense of actually having visited the department prior to start of RT; and 2) an information app. Three areas of information were available in the second app: 1) Q&As from the existing written information, presented both in writing and by a recorded voice; 2) practical information, such as maps, links to public transport options, telephone numbers and information about the patient hotel; and 3) a short, animated film about cancer. At this stage, the project management team was in contact with an advisory group at the RT-department to ensure the accuracy of the tool's content.

**Digital information tool version 1.0 pilot study (Step 3).** The relevance, comprehensiveness, feasibility, and usefulness of the digital tool version 1.0 was assessed in a pilot study, evaluated by individual telephone interviews with all participants. The pilot study included 11 participants (4 women and 7 men), aged between 57 and 75 years (mean 70 years). The participants were provided with an android smartphone and VR-glasses to use during the pilot project. Overall, most participants were positive to the experience of using the digital information tool. However, some found it difficult to get started with the tool as they had previously never used VR-glasses. Some participants found it challenging to learn how to operate a smartphone that was not the same type of mobile device they were used to. In addition, some patients expressed experiencing a sense of dizziness while using the VR-glasses.

Regarding the content of the information app, some patients requested additional information regarding specific preparatory procedures, e.g., about the x-ray prior to RT, while others asked for more pharmacological information. It became apparent that the current version 1.0

did not cover the subjects of dietary information to improve the effect of RT and healing of soft tissue. Other concerns important to the participants were specific side-effects from the RT, and technical details surrounding the whole RT procedure. The findings suggested there was a need for further clarification in the form of an extended step-by-step manual explaining how to set up and operate the digital information tool. Furthermore, during the pilot study, it became clear that the participants found that they utilized the information-app more frequently than the VR-app combined with the VR-glasses. Several participants appreciated the ability to revisit the information. Through the pilot study it became apparent that the possibility of a virtual visit to the RT-department reduced the patient's anxiety as it helped them to familiarize themselves with the environment. Several of the participants expressed that the digital information tool enabled them to share the information and visual experience of the RT-department with their family and friends, thus increasing their understanding and support. As for capturing the other side of the user experience, a presentation of the digital information tool 1.0 at a staff meeting for healthcare professionals generated creative discussion for the next stage of the development process.

## Prototyping

**Project management group & Innovation development team (Step 4).**   Valuable knowledge gained from the pilot study and the discussion at the staff meeting resulted in appropriate changes and advancements emerging in version 2.0. For example, the manual that includes information about how to set up and get started with the VR-glasses and mobile application was further improved to contain more information and photos of the VR-glasses. In response to the participants who voiced a preference to view the images on a mobile device instead of the VR-glasses due to motion sickness, an alternative version was created enabling the patient to experience the simulated environment but without the immersive element of the VR-glasses. The 2.0 version was updated to be accessible by both iPhone and Androids to enable the patient to use the applications on a mobile device they are comfortable with.

Parallel to the development of the mobile applications, an additional three short, animated films were produced through the co-design processes. The first film focuses on RT in general, the second film relates to RT specific to breast cancer, and the third film involves how physical exercise can help reduce fatigue during cancer treatment. The IDT, the project management group, the advisory groups, including participants from relevant specialties and the patient representative, were all engaged in the production of the films. Further, the films were co-produced with two regional cancer centers in Sweden, where consultation with the patient and family board at one of the regional cancer centers captured the patients' perspective.

The version 1.0 digital information tool contained images from only one of the two specific RT-departments providing RT in the current region for our project. Due to unforeseen circumstances in the form of extended waiting-times for the RT, some of the patients had to attend a second RT-department to receive their RT. This meant that the second RT-department in the region, situated in a different city, was also included in the project. To ensure that all patients received correct information and to enable their opportunity to view images from the specific RT-department in which they are sent to receive their RT, the production of 3D images from the second RT-department also had to be undertaken. The lessons learned from the first production of creating images were applied during the second production, as well as in the continuing discussions with the advisory group, healthcare staff, and the patient representative, which enabled a much smoother process.

**Stakeholders' interviews (Step 5).**   Two separate group interviews were conducted with healthcare professionals and team members from the IDT. The interview guide focused on

exploring stakeholders' experiences of co-designing a digital information tool along with any learning from their experience [35]. The interview began with an open-ended question; "Can you tell me about your experience working with the research team in co-designing a digital information tool?". The healthcare professionals shared reflections on their participation as stakeholders in the co-design process. They had enjoyed working with the research team and felt that their input and professional views had been valued and influential. Further, they accentuated that the experience was part of the stimulating process of developing the tool and they were eager for the digital information tool to be implemented in their RT services. Following the pilot study, the healthcare professionals saw that the digital information tool had potential for further developments to facilitate the coordination of services and scheduling to support task management and transport for patients, e.g., the ability to book a taxi. The members from the IDT had previous experience of working with researchers, despite the specific co-design process of including several stakeholders being new to them. They valued having been involved at an early stage of the process. Throughout the process, where communication with the research team had been interactive, their views had been listened to and the researchers had been able to contribute suggestions regarding the design and content. However, they did explain that, although they were aware of the benefits of involving different stakeholders in the co-design process, they noted challenges as digital technology is an evolving area advancing at a high speed.

## Discussion

Within the scope of this article, we provide a transparent description of the co-design process used to develop an innovative digital information tool employing PD methodology where several stakeholders participated as co-designers. Facilitating collaboration, structured PD activities can help researchers, healthcare professionals and patients to co-design patient information that meets the end users' needs [30, 32]. Co-designing patient information material is a complex endeavor and may involve challenges for both the research team and the stakeholders involved. The process requires that the goals, and objectives for each participant are clearly stated and agreed at the onset [36] and involves an appreciation of how the stakeholders take on diverse roles where each one brings key features into the project [11, 12].

Being aware of this, the project management group ensured they maintained an iterative partnership with the IDT and the stakeholders. Although discussions took place, no great conflicting interests emerged during the process. However, Smith, Wallengren [8] suggest that conflicts should not always be seen as something negative but as a source of development in the research process. Involving the IDT from the start was valuable and the collaboration proved to be essential in the development of both versions 1.0 and 2.0 of the digital information tools. Their expertise in driving innovation was combined with the expertise gained from the project management and advisory groups and, finally, the prerequisite fund of knowledge gained from the stakeholders.

By applying a person-centered approach, the project management group was encouraged to incorporate different dimensions within the tool to enable the patient to use it in line with their personal style of learning. The approach also included the families involved, as the patient effortlessly was able to share the digital tool with others. Some patients in the pilot study experienced a greater understanding and support from the family as they by using the digital tool gained more of an insight of living with cancer and RT. Additionally, from a person-centered perspective, this plays an imperative part, as the approach is not only limited to the patient, but also includes, as Santana, Manalili [18] and Ekman, Wolf [13] state, families and caregivers.

Engaging different stakeholders in the research process can improve the rigor, relevance, and reach of science (the 3 R's) [30, 36]. Employing co-design, where involved stakeholders

encouraged the researchers to broaden their understanding to look beyond traditional ways of producing patient information, enhanced the *rigor* of the process. Developing a high-quality digital information tool with different stakeholders, working in iterative steps throughout the process, is labor-intensive compared to more traditional approaches [8, 30, 31, 37]. However, it is important to remember that involving healthcare professionals and patients, who are the end users, from the onset of the project ensured the *relevance* of the project. Throughout, the dynamic perspectives of HL related to the patient's ability to adapt information in a stressful situation were also taken into consideration. Previous findings suggest that co-designed patient information creates material that most patients find relevant and may be considered to have higher levels of usability and preference compared to expert-designed patient information [4, 8, 32]. Involving patients and healthcare professionals in the co-design process and, additionally, working with the healthcare professionals in the implementation process, supports a wide *reach* of the digital information tool.

Consistent with previous studies [8, 38], we experienced that applying a co-design approach can be time-consuming. However, this approach is something that can generate both opportunities and challenges in the research process that involves technical elements such as digital technology that is constantly evolving at an exponential pace. In relation to the research process, in this article, the fast pace of advancement of the technology means that the specific digital technology applied in the project may not have been the first choice if the project had started today, as findings from the interview with the IDT suggested. However, although adopting to a co-design process may be time-consuming, the consequences of not involving stakeholders and not testing the digital information tool in a pilot study must be considered, as this may have a negative impact on the quality and usability of the end-product [8].

The positive effects of this innovative mode of providing information became apparent in the pilot study, where several of the participants highlighted the ability to revisit the information and the benefit of having a visual orientation of the RT-department. This led us to believe that there is a potential for the digital information tool to complement the one-to-one information clinical meetings as well as being a measure to improve HL, as supported by Mårtensson and Hensing [15] and eHealth literacy [24]. Williams, Blencowe [19] found that the healthcare professional at times found it challenging to explain the technical aspects of the RT environment and processes in such a way to avoid jargon and terminology unfamiliar to the patient. VR-technology has the potential to assist in processes at which previous traditional techniques fail [2, 21] enabling the patients a virtual visualize what is being explained. In a study by Johnson, Liszewski [29] 86% of the participants revealed that standard information did not fully cover the whole treatment process of RT. The participants did, however, see the benefits of being able of VR video viewing as a part of the preparation process prior start of RT treatment. The authors concluded a VR education tool has the potential to enhance standard patient education, increasing understanding of treatment and decrease anxiety. Similar results were found by Gao, Liu [28] who suggest that a virtual reality radiotherapy (VRRT) education program can have a positive effect in form of reducing anxiety and increasing the patient RT comprehension prior their initial RT session. In our case, the digital information tool, with its two separate apps, enables the patient to undergo a virtual experience of the technical environment through VR-technology, while the information app presents information in the form of text and animated films.

In line with the methodology, additional changes and further improvements were made following the pilot study from which the improved version 2.0 emerged. Other findings suggested that the mobile app containing information was utilized to a greater extent than the app with the VR-glasses. This was interesting, as, throughout the development process, the project management group and the IDT had believed the VR-glasses to be the most useful component of

the digital information tool. This hence reiterates the importance of involving end users in co-design as well as testing the acceptability, and feasibility of the tool in a pilot study. By involving patients as co-designers in the project, it was possible to identify obstacles in the form of unnecessarily complicated interfaces and by creating technological awareness.

There is a paucity in the literature reporting the adverse effects of VR-based interventions, however, caution should be observed, as VR is not without complications. From the pilot study, it became clear that motion sickness was a common VR-related symptom, a symptom also described by Baños, Espinoza [39]. This was viewed as being important information, with the result that the project management group and the IDT created a link in version 2.0 enabling the patient to view 3D images from the RT-departments directly on the mobile phone screen as a substitute to the VR-glasses, without the immersive VR-experience. Previous research suggests that there are positive effects of applying VR-technology in healthcare, where Zeng, Zhang [21] found significant improvement in cancer-related symptoms of fatigue. Further, they noted that VR-interventions had a positive effect on other cancer-related symptoms, such as anxiety, depression, pain, and cognitive dysfunction, although these findings were not statistically significant.

As mentioned, involving stakeholders in co-designing a digital information tool supports both the relevance and reach of the project. There are, however, still challenges ahead related to the implementation of the digital information tool, as failures or partial success are common in the implementation of technology-supported innovations [40]. Despite a growing interest and demand in the use of technological innovations and VR-based therapies within healthcare [21] and evidence highlighting benefits for both patients [6] and staff [41], simply imposing this digital technology on healthcare staff may not be sufficient for assisting them to improve the health and well-being pf the patients. Involving the healthcare professionals in co-designing and collaborating with them in the implementation process of the digital information tool enabled us to work from the bottom-up rather than top-down, which will not only improve the *reach* of the project, but it can also be a favorable factor that facilitates a successful implementation [40].

Greenhalgh, Wherton [40] suggest that there are several challenges to overcome before a digital information tool can be successfully implemented. They have produced a multi-level interdisciplinary framework including 6 domains (1. The Condition/Illness, 2. Technology, 3. Value proposition, 4. Adopter system, 5. Health/care organizations, and 6. Wider system). The 6 domains support implementation by identifying areas of complexity and mitigating risk at an early stage in the research process. In relation to the present article, we can relate foremost to domain 4. Greenhalgh, Wherton [40] point out challenges in relation to the intended adopters (domain 4) in the form of staff resistance to embracing new technology. By identifying intended adopters (patients and healthcare professionals) and employing a co-design process, we managed to address these types of challenges early in the project.

## Strengths and weaknesses

A strength of this article is the transparent description of the PD process presenting the involvement of patients, healthcare professionals and the IUD team as co-designers which ensured the 3 R's in the project. While something that can be viewed as a limitation is that the research team was unsuccessful in recruiting patients to the initial workshops. Even though the patient participants need of information was measured through a patient survey and complementary individual short briefings their lack of participation in the initial workshops may have resulted in missing valuable information regarding the patient's perspective of information needs related to RT treatment. What also may be seen as a limitation and risk for

introducing bias is that the patient representative was a member of the IDT. By being aware of this was addressed and discussed during the process.

## Future directions

We have demonstrated that involving stakeholders in co-designing a digital information tool with VR-technology enables the information to be tailored according to the patient's individual needs and style of learning. Further studies will take place to further explore the potential that the digital information tool holds in relation to increased HL and eHealth and the potential to enhance the sense of preparedness prior commencing RT.

The results from the pilot study were encouraging, and the information obtained was very useful in the further development of the digital tool to become version 2.0. The feasibility and usability of the digital information tool version 2.0 will be formally tested in a RCT with a robust research design and sample size to ensure high methodological quality. Further, in planned future trials and implementations, consideration will be given to the 6 domains of the multi-level interdisciplinary framework [40].

## Conclusion

In conclusion, the PD methodology, combined a protocol developed by Elwyn [33] was successfully applied in the development of the digital information tool. The needs and preferences of the patients and healthcare professionals were identified in the initial needs assessment (Step 1). Involving the end users enabled us to generate improved design solutions with clear interfaces and user-friendly applications. We believe that the modularity of the co-design approach and working with experts in gamification within healthcare in developing the digital information tool was beneficial and is to be recommended for future similar initiatives. What we can conclude from our research is that involving different stakeholders who are the end users throughout the research process, although time-consuming, is worthwhile, as this approach had a positive effect on the rigor, relevance, and reach of the project in form of development of the digital tool including relevant information material, the design and interfaces of the tool and the future implementation process.

## Acknowledgments

The authors would like to thank first and foremost the participating patients, healthcare professionals, advisory group and the IDT for their input, expertise, and interesting discussions. We would also like to thank Professor Thomas Björk-Eriksson director at Regional Cancer Centre West for support and Aileen Ireland for her excellent work with the language check. Finally, we would like to thank the Department of Nursing and IMPROVE, School of Health and Welfare, Jönköping University.

## Author Contributions

**Conceptualization:** Annika Grynne, Maria Browall, Sofi Fristedt, Karin Ahlberg, Frida Smith.

**Methodology:** Annika Grynne, Frida Smith.

**Project administration:** Annika Grynne, Maria Browall, Frida Smith.

**Supervision:** Maria Browall, Frida Smith.

**Writing – original draft:** Annika Grynne.

**Writing – review & editing:** Maria Browall, Sofi Fristedt, Frida Smith.

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
