## [Decision Letter · Decision Letter 0]

18 Jan 2021

PONE-D-20-33883

Integrating perspectives of patients, healthcare professionals, system developers and academics in the co-design of a digital information tool.

PLOS ONE

Dear Dr. Grynne,

Thank you for submitting your manuscript to PLOS ONE. After careful consideration, we feel that it has merit but does not fully meet PLOS ONE’s publication criteria as it currently stands. Therefore, we invite you to submit a revised version of the manuscript that addresses the points raised during the review process.

We look forward to receiving your revised manuscript.

Kind regards,

Alessandra Solari, M.D.

Academic Editor

PLOS ONE

Journal Requirements:

2. Please amend your Methods section to include all of the information about ethics approval and consent that you provided in the Ethics Statement.

Reviewers' comments:

Reviewer's Responses to Questions

**Comments to the Author**

1. Is the manuscript technically sound, and do the data support the conclusions?

Reviewer #1: Partly

Reviewer #2: Yes

2. Has the statistical analysis been performed appropriately and rigorously? 

Reviewer #1: N/A

Reviewer #2: I Don't Know

3. Have the authors made all data underlying the findings in their manuscript fully available?

Reviewer #1: No

Reviewer #2: Yes

4. Is the manuscript presented in an intelligible fashion and written in standard English?

Reviewer #1: Yes

Reviewer #2: Yes

5. Review Comments to the Author

Reviewer #1: The authors present a description of a participatory co-design development process of a “digital information tool” for patients with cancer awaiting radiotherapy consisting of different elements to be subsequently tested in a RCT. The paper is well written and easy to follow. In my view, this is surely relevant preparatory work that should be published to inform future researchers. However, the paper is mostly descriptive in nature and, although the headlines are there, lacks the expected structure of a scientific paper. Therefore, the paper needs major changes to clearly describe and justify the methods used and present the results and their implications. In the following, I will give some specific examples, following the outline of the manuscript.

(1) Abstract: Methods, results and discussion are missing. The abstract ends with the study question which should rather be at the beginning.

(2) Introduction: (a) The primary goal of the intervention is not clear as information, increased perception, health literacy and informed decisions are surely different aims. (b) The aim of the paper is given as “to describe the PD process”. This is not sufficient, as this does rather imply a study protocol than a scientific study.

(3) Methods: (a) Has the study protocol of the main study been published or registered? Please provide references, if applicable. (b) The chosen participatory design seems adequate, but there need to be more details especially concerning recruitment of participants (how, when, where from…) and rationale for sample sizes. It might be the case that all these have been pragmatic/convenience processes, but this should be stated. (c) The “Elwyn protocol” is given as one of the underlying approaches. First, I wonder if the term “Elwyn protocol” is appropriate, but more important, the framework cannot clearly be found here and as it addresses decision support interventions, does not clearly fit the proposed intervention here. (d) As stated above, under “Work Procedure”, important information about recruitment procedures are missing. Also, there are aspects in the results section e.g. on the conduction of the group interviews that rather belong here. For most parts of the study, methods are not reported (in detail), e.g. on the methods used for conducting the interviews and the qualitative analysis methods. Also, for the literature search and the “evidence synthesis”, methods are not reported. The procedure for identifying and analyzing “existing written information” also remains unclear. Also, the “pilot project” remains unclear as this seems to have been a controlled study, but there is no further information on this. The reported results therefore cannot be interpreted.

(4) Results: This section contains a lot of methods content (see above), which also highlights the lack of presented results.

(5) Discussion: (a) I acknowledge that the study aimed for a “transparent description of the co-design process…”, which is valuable, but can at present not be derived from the paper. Also, there are many aspects that seem to belong to the background or are repetitions (e.g. “Engaging different stakeholders in the research process can improve the rigor, relevance, and reach of science”) (l.320-321), which was clear from the outset. The same applies to several other paragraphs, e.g. just to state another one “However, although adopting to a co-design process may be time-consuming, the consequences of not involving stakeholders and not testing the digital information tool in a pilot project must be considered, as this will have a detrimental effect on the quality and usability of the end product.” This is nothing that was shown by this study. (b) In line 345 again the “positive effects” are mentioned and I am not sure what this refers to. (c) A section describing strength and limitation of this study is missing. For example, the fact that no patients participated in the initial workshops should be stated in this context as well as the (proposedly) convenience sampling. In contrast, only limitations of the developed interventions are stated, which is interesting, but of minor relevance for this paper.

(6) Conclusion: Here, you even refer to “…a greatly positive effect”, please clarify and present data or revise.

(7) Figure: (a) The figure is very helpful and I would suggest to structure both the methods and the results sections accordingly, using the same headlines as used in the figure. (b) Typo: RTC.

Reviewer #2: An interesting article that focuses on the importance of co-creating solutions for patients with patients, in this case a digital innovation for cancer patients.

It is unclear why the authors chose Spinuzzi as reference for co-design methodology, which is a bit old (2005) while the majority of patient engagement methodologies and best practices have been developed in the last five years.

No evidence that a review has been performed (even a quick one) of publications on patient involvement or co-design methodologies. No evidence of awareness of important patient engagement projects at european level like PARADIGM or PREFER, that, although focused on drug development and not innovative solution development, provide good Frameworks for patient engagement validated at multi-stakeholder level.

Governance: in the Project Management group no patients have been included, just researchers. Patient Experts should have been involved. The patient rep in the IDT group was also an employee, which could introduce bias.

Patients have not been involved in the review of existing written information.

Good point: involvement of patients in all steps of development, from design to evaluation.

6. PLOS authors have the option to publish the peer review history of their article (what does this mean?). If published, this will include your full peer review and any attached files.

Reviewer #1: No

Reviewer #2: No

---

## [Author Response · Author response to Decision Letter 0]

25 Feb 2021

Please see rebuttal letter that responds to each point raised by the academic editor and reviewer(s) uploaded as a separate file labeled 'Response to Reviewers' as advised in the Decision Letter.

---

## [Decision Letter · Decision Letter 1]

11 Apr 2021

PONE-D-20-33883R1

Integrating perspectives of patients, healthcare professionals, system developers and academics in the co-design of a digital information tool.

PLOS ONE

Dear Dr.Grynne,

Thank you for submitting your manuscript to PLOS ONE. After careful consideration, we feel that it has merit but does not fully meet PLOS ONE’s publication criteria as it currently stands. Therefore, we invite you to submit a revised version of the manuscript that addresses the points raised during the review process.

Specifically, I recommend that you revise the manuscript in response to the points raised by Reviewer 1.

Please submit your revised manuscript by May 1st. If you will need more time than this to complete your revisions, please reply to this message or contact the journal office at plosone@plos.org. Please include the following items when submitting your revised manuscript:

We look forward to receiving your revised manuscript.

Kind regards,

Alessandra Solari, M.D.

Academic Editor

PLOS ONE

Journal Requirements:

Reviewers' comments:

Reviewer's Responses to Questions

**Comments to the Author**

1. If the authors have adequately addressed your comments raised in a previous round of review and you feel that this manuscript is now acceptable for publication, you may indicate that here to bypass the “Comments to the Author” section, enter your conflict of interest statement in the “Confidential to Editor” section, and submit your "Accept" recommendation.

Reviewer #1: (No Response)

Reviewer #2: All comments have been addressed

2. Is the manuscript technically sound, and do the data support the conclusions?

Reviewer #1: Yes

Reviewer #2: Yes

3. Has the statistical analysis been performed appropriately and rigorously? 

Reviewer #1: N/A

Reviewer #2: I Don't Know

4. Have the authors made all data underlying the findings in their manuscript fully available?

Reviewer #1: Yes

Reviewer #2: (No Response)

5. Is the manuscript presented in an intelligible fashion and written in standard English?

Reviewer #1: Yes

Reviewer #2: (No Response)

6. Review Comments to the Author

Reviewer #1: The authors have revised the manuscript according to the reviewers‘ comments and I only have a few remaining comments.

(1) General: The wording in the newly added passages sometimes seems not completely adequate, but being not a native speaker myself, I would leave this to the copy editors.

(2) Abstract: In my view, the abstract is still not acceptable. To combine methods & results is neither adequate nor necessary. In addition, there is hardly any information about the PD and its results and implications. Instead, there is information about the digital tool (which is not the focus of the article) and some redundancy (e.g. “Involving…proved to be valuable” & “Involving…proved to be beneficial”). Please revise and clearly state study results.

(3) L.167: “PD is about the direct involement (sic!) of people in the co-design…” As co-design is about involving people, I think, this sentence does not make sense.

(4) L.385-7: At some instances, the authors tend to describe the result of the process rather than the process, e.g. “Our article is one of the first to present a digital information tool that includes VR-technology.” As the paper is not about the tool, but the process, this seems incorrect, please check.

(5) L.506: “A strength of this article…” This section should target the strength of the study, not the paper, please adapt.+

(6) L.508: The 3 R’s are mentioned (as such) for the first time here, please use this abbreviation above.

(7) L.542: Was there really no funding (including the RCT)?

(8) Figure: I am not completely sure what led to the changes in the figure, but they seem o.k., except the term “Prospective RCT study”, please use “Randomized controlled trial”.

Reviewer #2: (No Response)

7. PLOS authors have the option to publish the peer review history of their article (what does this mean?). If published, this will include your full peer review and any attached files.

Reviewer #1: No

Reviewer #2: No

---

## [Author Response · Author response to Decision Letter 1]

23 Apr 2021

Please see uploaded document 'Respons to Reviewers'

Journal Requirements: Please review your reference list to ensure that it is co

mplete and correct. 

Response: Reference Fristedt et al 2021 has been successfully accepted and published. The reference list has been updated accordingly. 

Response to reviewer 1:

1. Thank you for highlighting this. The text has been checked again and hopefully reads better now. 

2. You have got a valuable point; I have now amended the text in the abstract to include the PD more specifically. Please see revised text. 

I very much appreciate your view; however, I still would like to argue that this is original research that is more descriptive of a kind hence the method and findings are mixed. 

3. You are of course correct about this. I have now amended the text. 

4. Absolutely, I agree with you that the article is about “the process”. I have removed the first sentence in the discussion accordingly and it now reads as you can see in the next column. 

5. You are of course right, and I agree that is the way to write in a traditional scientific study. This however, although being original research is not a traditional scientific study, we do not aim to test something here, we aim to describe the PD approach with co-design.

6. To seek to clarify for the reader, I have now amended the text where I am initially writing about the rigor, reach and relevance of science and put in the 3 R’s in brackets. 

I have also amended the text according to your suggestion – thank you very much. 

7. I apologise for being unclear with regards to funding. 

This article is part of my PhD. The project, development of the digital tool and RCT has been received from Regional Cancer Centre West (RCC), Chalmers innovation office, Knut and Ragnvi Jacobsson Family Foundation and Jönköping University, IMPROVE. 

8. Ah, there are no great changes other that I after you previously pointed out my spelling mistake on RCT in your first review where I had written RTC. 

Randomized controlled trial reads better than “prospective RCT study” thank you.

---

## [Decision Letter · Decision Letter 2]

7 Jun 2021

Integrating perspectives of patients, healthcare professionals, system developers and academics in the co-design of a digital information tool.

PONE-D-20-33883R2

Dear Dr. Grynne,

We’re pleased to inform you that your manuscript has been judged scientifically suitable for publication and will be formally accepted for publication once it meets all outstanding technical requirements.

Kind regards,

Frédéric Denis, Ph.D.

Academic Editor

PLOS ONE

Additional Editor Comments (optional):

Reviewers' comments:

Reviewer's Responses to Questions

**Comments to the Author**

1. If the authors have adequately addressed your comments raised in a previous round of review and you feel that this manuscript is now acceptable for publication, you may indicate that here to bypass the “Comments to the Author” section, enter your conflict of interest statement in the “Confidential to Editor” section, and submit your "Accept" recommendation.

Reviewer #1: All comments have been addressed

Reviewer #2: All comments have been addressed

2. Is the manuscript technically sound, and do the data support the conclusions?

Reviewer #1: (No Response)

Reviewer #2: Yes

3. Has the statistical analysis been performed appropriately and rigorously? 

Reviewer #1: (No Response)

Reviewer #2: I Don't Know

4. Have the authors made all data underlying the findings in their manuscript fully available?

Reviewer #1: (No Response)

Reviewer #2: Yes

5. Is the manuscript presented in an intelligible fashion and written in standard English?

Reviewer #1: (No Response)

Reviewer #2: Yes

6. Review Comments to the Author

Reviewer #1: (No Response)

Reviewer #2: (No Response)

7. PLOS authors have the option to publish the peer review history of their article (what does this mean?). If published, this will include your full peer review and any attached files.

Reviewer #1: No

Reviewer #2: No

---

## [Editor Report · Acceptance letter]

1 Jul 2021

PONE-D-20-33883R2 

Integrating perspectives of patients, healthcare professionals, system developers and academics in the co-design of a digital information tool 

Dear Dr. Grynne:

I'm pleased to inform you that your manuscript has been deemed suitable for publication in PLOS ONE. Congratulations! Your manuscript is now with our production department. 

Kind regards, 

on behalf of

Dr. Frédéric Denis 

Academic Editor

PLOS ONE